# Effect of Sodium Glucose Cotransporter 2 Inhibitors on Renal Function in Patients with Nonalcoholic Fatty Liver Disease and Type 2 Diabetes in Japan

**DOI:** 10.3390/diagnostics10020086

**Published:** 2020-02-06

**Authors:** Kota Yano, Yuya Seko, Aya Takahashi, Shinya Okishio, Seita Kataoka, Masashi Takemura, Keiichiroh Okuda, Naoki Mizuno, Hiroyoshi Taketani, Atsushi Umemura, Taichiro Nishikawa, Kanji Yamaguchi, Michihisa Moriguchi, Takeshi Okanoue, Yoshito Itoh

**Affiliations:** 1Department of Molecular Gastroenterology and Hepatology, Kyoto Prefectural University of Medicine, Kyoto 6028566 Japan; yanokota@koto.kpu-m.ac.jp (K.Y.); ayataka@koto.kpu-m.ac.jp (A.T.); okishin@koto.kpu-m.ac.jp (S.O.); s1120@koto.kpu-m.ac.jp (S.K.); mtakem@koto.kpu-m.ac.jp (M.T.); k-okuda@koto.kpu-m.ac.jp (K.O.); naoban@koto.kpu-m.ac.jp (N.M.); take1012@koto.kpu-m.ac.jp (H.T.); aumemura@koto.kpu-m.ac.jp (A.U.); taichi@koto.kpu-m.ac.jp (T.N.); ykanji@koto.kpu-m.ac.jp (K.Y.); mmori@koto.kpu-m.ac.jp (M.M.); yitoh@koto.kpu-m.ac.jp (Y.I.); 2Department of Gastroenterology and Hepatology, Saiseikai Suita Hospital, Osaka 5640013, Japan; okanoue@suita.saiseikai.or.jp

**Keywords:** sodium-glucose cotransporter-2 inhibitor, nonalcoholic fatty liver disease, chronic kidney disease, type 2 diabetes

## Abstract

Sodium-glucose cotransporter-2 inhibitors (SGLT2I) have been reported to have renal-protective effects in patients with type 2 diabetes (T2DM). This a retrospective study aimed to evaluate the effect of SGLT2I on renal function in patients with nonalcoholic fatty liver disease (NAFLD) and T2DM. We analyzed 69 consecutive patients with a biopsy-proven NAFLD and T2DM with an estimated glomerular filtration rate (eGFR) >60 mL/min. Of these 69 patients, 22 received SGLT2I and 47 were treated without SGLT2I. Liver function and eGFR were analyzed at baseline and after three years. Body mass index, liver function and HbA1c improved significantly in both groups. In the total population, the median eGFR declined from 80.7 mL/min at the baseline to 74.9 mL/min at the end of follow-up. The median eGFR at the baseline/end of follow-up was 81.2/80.4 mL/min in patients treated with SGLT2I and 80.2/70.8 mL/min in patients treated without SGLT2I. Multivariate analysis identified an increased FIB-4 index with an odds ratio (OR) of 4.721, (*p* = 0.045) and SGLT2I treatment (OR 0.263, *p* = 0.033) as predictive factors for decreased eGFR. SGLT2I treatment has a protective effect on the renal function for NAFLD with T2DM. A long-term, randomized, controlled trial is warranted to confirm the renal protective effect of SGLT2I in NAFLD patients with T2DM.

## 1. Introduction

Nonalcoholic fatty liver disease (NAFLD) is the most common chronic liver disease in Japan, and it affects up to 25–30% of the general adult population worldwide [1]. NAFLD is known to be associated with extrahepatic diseases, such as type 2 diabetes mellitus (T2DM), cardiovascular (CV) disease and chronic kidney disease (CKD) [2,3,4]. The association between NAFLD and CKD has attracted considerable attention, and many studies support the concept that NAFLD affects the incidence of CKD [5,6]. Patients with NAFLD have a high risk of incident CKD, which is considered to be associated with the severity of the NAFLD. CKD is also one of the most common complications of T2DM. Thus, among patients with NAFLD, the presence of T2DM increases the risk of the development of CKD [7,8]. Currently, there are no established pharmacotherapies for NAFLD in patients with T2DM. However, antidiabetic agents, including pioglitazone [9,10] and glucagon-like peptide-1 (GLP-1) agonists [11], are expected to reduce hepatic fat content or hepatic inflammation in NAFLD. Several studies have reported the efficacy of sodium-glucose cotransporter-2 inhibitors (SGLT2I) in NAFLD with T2DM [12,13]. Patients with T2DM and high CV risk that received treatment with SGLT2I showed a slower progression of CKD and lower rates of renal events compared to patients receiving a placebo treatment [14,15]. There is a lack of longitudinal follow-up data regarding changes in renal function in NAFLD patients with T2DM. The renoprotective effect of SGLT2I in NAFLD remains unclear. This study aims to clarify the association between liver function and renal function in the follow-up period and the impact of SGLT2I treatment on renal function in NAFLD patients with T2DM.

## 2. Results

### 2.1. Patient Characteristics

A total of 69 patients with NAFLD and T2DM with estimated glomerular filtration rate (eGFR) >60 mL/min were analyzed. Table 1 summarizes the demographic profiles and the laboratory and histologic data of the study patients at baseline. The median age was 58 years, 33 patients (47.8%) were men, 58 patients (84.1%) were diagnosed with nonalcoholic steatohepatitis (NASH), 38 (55.1%) had hypertension and 50 (72.5%) had hyperlipidemia. Eight patients had cirrhosis. Among the 22 patients in the SGLT2I group, dapagliflozin was the most prescribed SGLT2I (*n* = 10), followed by canagliflozin (*n* = 7), ipragliflozin (*n* = 3) and empagliflozin (*n* = 2). There was no significant difference between HbA1c, aspartate aminotransferase (AST) and alanine aminotransferase (ALT), at baseline, according to SGLT2I treatment. The BMI of the SGLT2I group (29.3 kg/m^2^) was significantly greater than that of the nonSGLT2I group (26.2 kg/m^2^, *p* = 0.007). The median eGFR of patients with and without SGLT2I treatment at baseline was 81.2 mL/min and 80.2 mL/min, respectively (*p* = 0.187).

### 2.2. Changes in Biochemical Results during the Follow-Up Period

Median serum levels of AST, ALT, GGT, total cholesterol, HbA1c, and Type IV collagen 7s and the median FIB-4 index decreased significantly during the follow-up period in patients with and without SGLT2I treatment, as shown in Table 2; BMI also significantly reduced in both groups. There was a significant decrease in eGFR from 80.7 mL/min at baseline to 74.9 mL/min at 3 years after treatment in the overall population (*p* < 0.001). In the SGLT2I group, the median eGFR decreased from 81.2 mL/min at baseline to 80.4 mL/min 3 years later (*p* = 0.077), as shown in Table 2a. In contrast, the median eGFR in the nonSGLT2I group decreased significantly from 80.2 mL/min at baseline to 70.8 mL/min 3 years later (*p* < 0.001), as shown in Table 2b and Figure 1.

Abbreviations are defined in Table 1.

### 2.3. Factors Associated with Decreased eGFR

Among the 69 patients with NAFLD and T2DM, 48 (69.6%) had a decrease from baseline in eGFR and 21 (30.4%) had an increase from baseline in eGFR after 3 years. With the exception of gender, baseline characteristics did not differ between the groups. The change in the FIB-4 index tended to be higher among patients with increased eGFR (*p* = 0.087). The prevalence of patients with SGLT2I treatment tended to be greater among patients with increased eGFR (47.6%) than among patients with decreased eGFR (25.0%, *p* = 0.092). We performed multivariate analysis using sex, age, systolic blood pressure at baseline, change in FIB-4 index, and SGLT2I treatment as factors, as shown in Table 3. The analysis identified an increased FIB-4 index with an odds ratio (OR) of 4.72 (*p* = 0.045) and with SGLT2I treatment (OR 0.263, *p* = 0.033) as risk factors for decreased eGFR. The median change in eGFR of patients with an increased FIB-4 index (−14.7 mL/min) was significantly greater than that of patients with a decreased FIB-4 index (−2.4 mL/min; *p* = 0.01). Among the 47 patients who were not treated with SGLT2I, 36 patients (76.6%) had a decreased eGFR, while 12 of the 22 patients (54.5%) who were treated with SGLT2I had a decreased eGFR 3 years later.

## 3. Discussion

In the present study, we investigated the effect of SGLT2I on renal function in patients with NAFLD. An increase in the FIB-4 index was a risk factor for decreasing eGFR, and SGLT2I treatment appeared to slow CKD progression among Japanese patients with NAFLD and T2DM. This study was the first to clarify the relationship between changes in the FIB-4 index and renal function and the impact of SGLT2I on renal function in Japanese patients with NAFLD, based on the results of liver biopsies.

The risk of incident CKD was greater in patients with NAFLD than without, and several studies have reported that risk was associated with the severity of NAFLD [5,15]. The pathophysiological links between the FIB-4 index and renal function remain unclear, however. NAFLD, especially when linked to an increased FIB-4 index per sé and the release of inflammatory, thrombogenic, oxidative, vasoactive and fibrogenic mediators by adipose tissue, may lead to the development and progression of CKD [16,17].

The renal benefits of SGLT2I for patients with T2DM were demonstrated in several large randomized controlled trials (RCT) [14,15]. Treatment with empagliflozin was associated with a slower progression of CKD and lower rates of clinical renal events, such as incident nephropathy, increasing serum creatinine levels and frequency of renal replacement therapy [14]. Another study reported that treatment of T2DM patients with canagliflozin was associated with a greater reduction in albuminuria and the composite outcomes of a sustained 40% reduction in eGFR, the need for renal replacement therapy, or death from renal causes [15]. However, the participants in these studies were limited to patients with high CV risk. In this study, our results showed that SGLT2I had the same renoprotective effects in patients with NAFLD and without high CV risk. The mechanism underlying the beneficial effects of SGLT2I on renal function is believed to be based on several pathways. The reduction in the renal absorption of glucose and sodium in the proximal tubules leads to reduced intraglomerular hyperfiltration. In addition, lowering sodium and volume overload are thought to reduce renal burden as well as glucotoxicity in the kidney. The inhibition of the Na^+^/H^+^ exchanger 3 (NHE3) membrane protein, which is coexpressed with SGLT2 in the early proximal tubule, produces natriuresis and is also associated with renal benefits [18].

This study had several limitations. It was a retrospective, single-center study, and the number of subjects was not sufficient to confirm the results. An RCT would be the best way to address the potential benefits of SGLT2I, and further RCT using a larger number of subjects will be required to draw firm conclusions. Determination of the long-term effect of SGLT2I treatment on renal function and CV events in patients with NAFLD is also required. Furthermore, we did not collect information on proteinuria, and we used a creatinine-based equation to estimate eGFR, a calculation that may not be accurate in patients with cirrhosis.

In conclusion, change in the FIB-4 index was associated with renal function, and treatment with SGLT2I was shown to have a renoprotective effect in NAFLD patients with T2DM. Considering their effects on the protection of liver and renal function, SGLT2I may be candidates as an appropriate therapy for NAFLD patients with T2DM. Prospective RCTs are required to confirm our results and establish treatment strategies for patients with NAFLD and T2DM.

## 4. Materials and Methods

### 4.1. Patients

A total of 69 biopsy-proven Japanese NAFLD patients with T2DM were enrolled in this retrospective study. All patients had eGFR > 60 mL/min at baseline and were followed up for more than 3 years. Patients with other liver diseases, consuming more than 20 g of alcohol per day, or those with evidence of decompensated liver cirrhosis or hepatocellular carcinoma were excluded from this study. Of the 69 patients, 22 received SGLT2I (canagliflozin 100 mg or ipragliflozin 50 mg) within 3 months after liver biopsy. All patients provided written informed consent, and the study was conducted in accordance with the 2013 version of the Declaration of Helsinki. The study protocol was approved by the institution’s human research committees (ERB-C-544-2).

### 4.2. Laboratory and Clinical Parameters

Venous blood samples were collected in the morning after a 12 h overnight fast. Laboratory assays included blood cell counts and measurements of serum concentrations of aspartate aminotransferase (AST), alanine aminotransferase (ALT), gamma-glutamyl transpeptidase (GGT), total cholesterol, triglycerides, fasting plasma glucose (FPG) and type IV collagen 7s. Hemoglobin A1c (HbA1c) was assayed using high-performance liquid chromatography and was expressed as National Glycohemoglobin Standardization Program (NGSP) units. These parameters were measured with standard clinical chemistry laboratory techniques. Body mass index (BMI) was calculated as weight in kilograms/(height in meters^2^). T2DM was diagnosed according to the Report of the Expert Committee on the Diagnosis and Classification of Diabetes Mellitus or based on the administration of antiT2DM agents. Patients with serum cholesterol concentrations > 220 mg/dl or triglyceride concentrations > 160 mg/dl, or who were receiving treatment with anti-dyslipidemia agents, were defined as having dyslipidemia. The Fibrosis-4 (FIB-4) index was calculated as follows: ([age (years) × AST (U/L)]/platelet count [109/L]) × (ALT [U/L])1/2. The following formulae for Japanese individuals were used to calculate eGFR by gender [6]: for males, eGFR (mL/min/1.73 m^2^) = 194 × [age] − 0.287 × [serum creatinine (mg/dL)] − 1.094; for females, eGFR (mL/min/1.73 m^2^) = 194 × [age] − 0.287 × [serum creatinine (mg/dL)] − 1.094 × 0.739.

### 4.3. Liver Histology

All enrolled patients underwent a percutaneous liver biopsy under ultrasonic guidance. The liver specimens were embedded in paraffin and stained with hematoxylin, eosin and Masson–trichrome. The specimens were evaluated by two hepatic pathologists who were blinded to the clinical findings. An adequate liver biopsy sample was defined as a specimen >1.5 cm long and/or having more than 11 portal tracts. NASH was defined as steatosis with lobular inflammation and ballooning degeneration, with or without Mallory—Denk bodies or fibrosis. Patients with liver biopsy specimens that showed simple steatosis or steatosis with nonspecific inflammation were diagnosed with NAFL. Specimens with steatosis of <5%, 5–33%, >33–66%, or >66% were scored as steatosis grades 0, 1, 2 and 3, respectively. For mild, moderate and severe ballooning and inflammation (acinar and portal), the necroinflammatory grades were 1, 2 and 3, respectively. The severity of hepatic fibrosis (stage) was scored as stage 1, zone 3 perisinusoidal fibrosis; stage 2, zone 3 perisinusoidal fibrosis with portal fibrosis; stage 3, zone 3 perisinusoidal fibrosis and portal fibrosis with bridging fibrosis; and stage 4, cirrhosis.

### 4.4. Statistical Analysis

The distribution of subject characteristics was assessed by the chi-square test or Mann–Whitney’s U test, as appropriate. We performed logistic regression analysis to evaluate predictive factors of deterioration of eGFR adjusted for sex (male, female), age (<60 years, ≥60 years), change in FIB-4 index (yes, no) and SGLT2I medication (yes, no). All reported *p* values were two-sided, and the significance level was set at 0.05. Statistical comparisons were performed with SPSS software (SPSS Inc., Chicago, IL, USA).

## Figures and Tables

**Figure 1 diagnostics-10-00086-f001:**
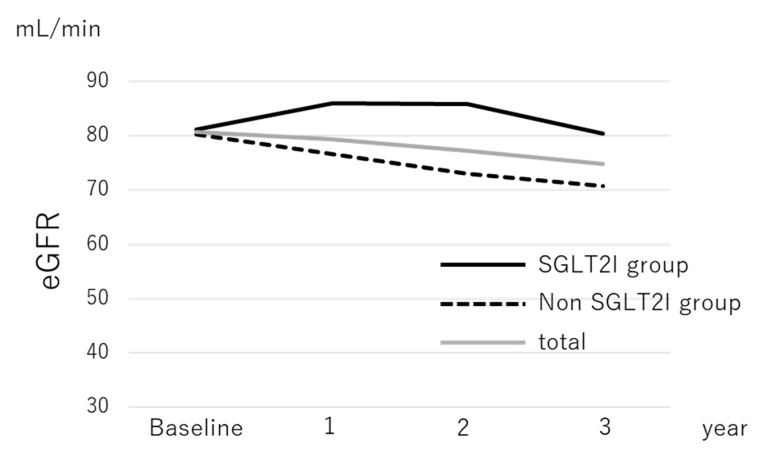
The change in the median estimated glomerular filtration rate of patients with nonalcoholic fatty liver disease according to sodium-glucose cotransporter-2 inhibitor therapy.

**Table 1 diagnostics-10-00086-t001:** The baseline characteristics of 69 patients with nonalcoholic fatty liver disease and diabetic mellitus treated with and without a sodium-glucose cotransporter-2 inhibitor (SGLT2).

Variable	Total (*n* = 69)	SGLT2I Group (*n* = 22)	NonSGLT2I Group (*n* = 47)	*p* Value
NASH	58 (84.1%)	21 (95.4%)	37 (78.7%)	0.092
Hypertension	38 (55.1%)	14 (63.6%)	24 (48.9%)	0.437
Hyperlipidemia	50 (72.5%)	15 (68.2%)	35 (74.5%)	0.579
Sex, male	33 (47.8%)	9 (40.9%)	24 (51.0%)	0.452
Age, years	58 (41–79)	56 (41–78)	59 (41–79)	0.388
BMI, kg/m^2^	27.9 (17.9–39.0)	29.3 (24.6–38.8)	26.2 (17.9–39.0)	0.007
Albumin, g/dL	4.4 (3.4–5.2)	4.4 (3.5–5.2)	4.4 (3.4–5.2)	0.995
AST, IU/L	45 (12–161)	46 (12–133)	45 (13–161)	0.802
ALT, IU/L	60 (10–242)	55.5 (18–237)	61 (10–242)	0.344
GGT, IU/L	72 (14–263)	78.5 (28–222)	72 (14–263)	0.567
Platelet count, × 10^3^/μL	199 (99–457)	199 (108–457)	199 (99–382)	0.985
Total cholesterol, mg/dL	189 (127–304)	184 (127–304)	192.5 (133–265)	0.568
Triglycerides, mg/dL	151 (52–739)	146 (52–327)	153 (55–739)	0.364
LDL-C, mg/dL	115 (66–195)	113.5 (75–195)	115.5 (66–177)	0.545
HDL-C, mg/dL	48 (26–99)	53 (36–99)	48 (26–84)	0.212
FPG, mg/dL	127.5 (61–325)	115 (88–191)	134 (61–325)	0.349
HbA1c, %	7.0 (5.5–11.0)	7.1 (5.6–10.4)	6.9 (5.5–11.0)	0.797
FIB-4 index	1.86 (0.27–7.84)	1.95 (0.27–4.14)	1.68 (0.61–7.84)	0.887
eGFR, mL/min/1.73 m^2^	80.7 (61.6–145.0)	81.2 (67.2–145.0)	80.2 (61.6–116.6)	0.187
Type IV collagen 7s, ng/mL	5.4 (2.8–12.0)	6.3 (3.8–9.0)	5.0 (2.8–12.0)	0.164
Fibrosis stage (0/1/2/3/4)	16/20/20/5/8	3/6/9/1/3	13/14/11/4/5	0.504
Steatosis (1/2/3)	14/43/12	4/16/2	10/27/10	0.384
Inflammation (1/2/3)	33/30/6	6/14/2	27/16/4	0.053
Ballooning (0/1/2)	11/26/32	1/8/13	10/18/19	0.152

Results are presented as *n* (%) for qualitative data or as median (range) for quantitative data. Abbreviations: SGLT2I, sodium-glucose cotransporter-2 inhibitor; NASH, nonalcoholic steatohepatitis; T2DM, type 2 diabetes mellitus; BMI, body mass index; AST, aspartate aminotransferase; ALT, alanine aminotransferase; GGT, gamma-glutamyl transferase; LDL-C, low-density lipoprotein cholesterol; HDL-C, high-density lipoprotein cholesterol; FPG; fasting plasma glucose; FIB-4, fibrosis-4; eGFR; estimated glomerular filtration rate.

**Table 2 diagnostics-10-00086-t002:** The characteristics of patients with nonalcoholic fatty liver disease in **(a)** the SGLT2I group and **(b)** the nonSGLT2I group, at baseline and after ≥ 3 years of follow-up.

Variable	Baseline	After ≥ 3 Years	*p* Value
**(a)** **the SGLT2I group**			
BMI, kg/m^2^	29.3 (24.6–38.8)	28.9 (23.3–38.5)	0.030
Albumin, g/dL	4.4 (3.5–5.2)	4.4 (4.0–5.0)	0.865
AST, IU/L	46 (12–133)	38.5 (15–70)	0.010
ALT, IU/L	55.5 (18–237)	39.5 (14–121)	0.007
GGT, IU/L	78.5 (28–222)	46 (16–189)	0.006
Platelet count, ×10^3^/μL	199 (108–457)	195 (117–381)	0.897
Total cholesterol, mg/dL	184 (127–304)	181 (109–256)	0.434
Triglycerides, mg/dL	146 (52–327)	133.5 (46–223)	0.846
LDL-C, mg/dL	113.5 (75–195)	100.5 (50–171)	0.877
HDL-C, mg/dL	53 (36–99)	48 (35–94)	0.983
FPG, mg/dL	115 (88–191)	137.5 (86–233)	0.154
HbA1c, %	7.1 (5.6–10.4)	7.1 (5.7–10.4)	0.543
FIB-4 index	1.95 (0.27–4.14)	1.83 (0.43–3.07)	0.095
eGFR, mL/min/1.73 m^2^	81.2 (67.2–145.0)	80.4 (50.2–121.8)	0.077
Type IV collagen 7s, ng/mL	6.3 (3.8–9.0)	5.3 (4.4–6.8)	0.006
**(b) the nonSGLT2I group**			
BMI, kg/m^2^	26.2 (17.9–39.0)	25.6 (18.4–36.7)	0.004
Albumin, g/dL	4.4 (3.4–5.2)	4.4 (2.8–5.0)	0.008
AST, IU/L	45 (13–161)	27 (12–83)	<0.001
ALT, IU/L	61 (10–242)	31 (12–136)	<0.001
GGT, IU/L	72 (14–263)	37 (14–171)	<0.001
Platelet count, ×10^3^/μL	199 (99–382)	213 (88–416)	0.013
Total cholesterol, mg/dL	189 (127–304)	184 (135–273)	0.015
Triglycerides, mg/dL	153 (55–739)	131.5 (53–574)	0.053
LDL-C, mg/dL	115.5 (66–177)	107 (55–185)	0.044
HDL-C, mg/dL	48 (26–84)	45 (22–87)	0.661
FPG, mg/dL	134 (61–325)	133 (86–349)	0.668
HbA1c, %	6.9 (5.5–11.0)	6.6 (5.5–11.1)	0.015
FIB-4 index	1.68 (0.61–7.84)	1.51 (0.39–6.88)	0.005
eGFR, mL/min/1.73 m^2^	80.2 (61.6–116.6)	70.8 (31.0–107.3)	<0.001
Type IV collagen 7s, ng/mL	5.0 (2.8–12.0)	4.9 (2.8–11.0)	0.010

**Table 3 diagnostics-10-00086-t003:** The factors associated with decreasing the estimated glomerular filtration rate in patients with nonalcoholic fatty liver disease by multivariate analysis.

Variable	Category	OR (95% CI) ^a^	*p* Value
Sex	1: men		0.093
	2: female		
Age, yr	Per 1 yr		0.366
Systolic blood pressure	Per 1 mmHg		0.943
Change in FIB-4 index	1: decrease	4.721 (1.036–21.510)	0.045
	2: increase		
SGLT2I treatment	1: no	0.263 (0.077–0.900)	0.033
	2: yes		

Abbreviations are defined in Table 1. OR: odds ratio, CI: confidence interval. ^a^ Estimated using logistic regression analysis.

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
