# Peer review of "Effect of Sodium Glucose Cotransporter 2 Inhibitors on Renal Function in Patients with Nonalcoholic Fatty Liver Disease and Type 2 Diabetes in Japan"

_diagnostics, 2020, doi:10.3390/diagnostics10020086_

Round 1

Reviewer 1 Report

The aims of this study are to clarify the association between liver function and renal function in the follow-up period and the impact of SGLT2I treatment on renal function in NAFLD patients with T2DM. Multivariate analysis identified increased FIB-4 index (odds ratio [OR] 4.574, P=0.049) and SGLT2I treatment (OR 0.290, 24 P=0.046) as predictive factors for decreased eGFR.

Comments

What was the primary endpoint in your study? I think changes in eGFR should be a primary endpoint in this study. How did you calculate the sample size needed? Please include blood pressure or the class of anti-hypertensive drugs as a factor in the multivariate analysis for decreased eGFR. Please show changes in parameters in patients treated with or without SGLT2i separately, instead of the original Table 2.

Author Response

Response to REVIEWER 1 Comments

Point 1: What was the primary endpoint in your study? I think changes in eGFR should be a primary endpoint in this study. How did you calculate the sample size needed? Please include blood pressure or the class of anti-hypertensive drugs as a factor in the multivariate analysis for decreased eGFR. Please show changes in parameters in patients treated with or without SGLT2i separately, instead of the original Table 2.

Response 1: As reviewer pointed, the primary endpoint of this retrospective study was the change in eGFR in patients with NAFLD and T2DM. However, because it was a retrospective study, the exact sample size to perform multivariate analysis is not confirmed. We added the small sample size as limitation in the “Discussion” section. (Page5, line 133)

We modified Table 2, showed changes in parameters in SGLT2I group and Non SGLT2I group.

We added the systolic blood pressure at baseline as a factor in multivariate analysis in Table 3. After adjusted with systolic blood pressure, increased FIB-4 index and treatment with SGLT2I were still identified as independent risk factor for decreased eGFR. (Table3)

Reviewer 2 Report

This is an observational study of patients with concomitant nonalcoholic fatty liver disease and type 2 diabetes. 25 to 30% happened to be treated with SGLT2 inhibitors and the remainder were not. The authors correctly pointed out that a randomized trial would be the best way to address potential benefit in this clinical situation

Author Response

Response to REVIEWER 2 Comments

Point 1: This is an observational study of patients with concomitant nonalcoholic fatty liver disease and type 2 diabetes. 25 to 30% happened to be treated with SGLT2 inhibitors and the remainder were not. The authors correctly pointed out that a randomized trial would be the best way to address potential benefit in this clinical situation

Response 1: In accordance with reviewer’s suggestion, we added that the randomized trial is needed to confirm the result in “Discussion” section. (page5, line 134)